# PMSFF: Improved Protein Binding Residues Prediction through Multi-Scale Sequence-Based Feature Fusion Strategy

**DOI:** 10.3390/biom14101220

**Published:** 2024-09-27

**Authors:** Yuguang Li, Xiaofei Nan, Shoutao Zhang, Qinglei Zhou, Shuai Lu, Zhen Tian

**Affiliations:** 1School of Computer and Artificial Intelligence, Zhengzhou University, Zhengzhou 450001, China; liyuguang@gs.zzu.edu.cn (Y.L.); iexfnan@zzu.edu.cn (X.N.); ieqlzhou@zzu.edu.cn (Q.Z.); 2School of Life Sciences, Zhengzhou University, Zhengzhou 450001, China; zhangst@zzu.edu.cn; 3Longhu Laboratory of Advanced Immunology, Zhengzhou 450001, China; 4National Supercomputing Center in Zhengzhou, Zhengzhou University, Zhengzhou 450001, China; 5Yangtze Delta Region Institute (Quzhou), University of Electronic Science and Technology of China, Quzhou 324003, China

**Keywords:** protein binding residues, sequence-based feature, attention mechanism, deep learning

## Abstract

Proteins perform different biological functions through binding with various molecules which are mediated by a few key residues and accurate prediction of such protein binding residues (PBRs) is crucial for understanding cellular processes and for designing new drugs. Many computational prediction approaches have been proposed to identify PBRs with sequence-based features. However, these approaches face two main challenges: (1) these methods only concatenate residue feature vectors with a simple sliding window strategy, and (2) it is challenging to find a uniform sliding window size suitable for learning embeddings across different types of PBRs. In this study, we propose one novel framework that could apply multiple types of PBRs **P**rediciton task through **M**ulti-scale **S**equence-based **F**eature **F**usion (PMSFF) strategy. Firstly, PMSFF employs a pre-trained language model named ProtT5, to encode amino acid residues in protein sequences. Then, it generates multi-scale residue embeddings by applying multi-size windows to capture effective neighboring residues and multi-size kernels to learn information across different scales. Additionally, the proposed model treats protein sequences as sentences, employing a bidirectional GRU to learn global context. We also collect benchmark datasets encompassing various PBRs types and evaluate our PMSFF approach to these datasets. Compared with state-of-the-art methods, PMSFF demonstrates superior performance on most PBRs prediction tasks.

## 1. Introduction

Multitudinous cellular functions, such as immune reaction, signal transduction, and transcription are driven by protein interactions [1,2]. Those interactions are carried out with a variety of other molecules including DNA, RNA, small ligands and other proteins [3,4]. Research has demonstrated that antibody-antigen interaction plays a crucial role in the immune response and is a specific type of protein-protein interaction [5]. Consequently, accurate detecting of protein interactions contributes to the construction of protein interaction networks [6,7,8], annotation of protein functions [9,10], understanding of molecular mechanisms about diseases [11], design of new vaccines [12,13] and development of novel therapeutic antibodies [14].

Following the success of the Human Genome Project, millions of protein sequences have been determined and are stored in public databases [15,16]. Determining protein binding residues through traditional biological experiments is time-consuming and costly [17]. Therefore, a variety of computational methods have been proposed in recent years to address this issue. Computational methods for predicting protein binding residues (PBRs) can be broadly classified into sequence-based and structure-based approaches [18]. Sequenced-based methods have a broad application range because structure-based methods rely on available protein structures. With the development of protein structure prediction such as AlphaFold [19] and AlphaFold DB [20], structure-based methods are improved by much more predicted protein structures [21,22]. Although structure-based computational methods can provide better results, we focus on sequence-based methods in this study as they are usually much faster and cheaper. PBRs encompass various interaction types, including protein-protein, protein-nucleotide, protein-small ligand, and antibody-antigen binding residues [2,4,23]. As summarized in Table 1, a diverse range of machine learning and deep learning models, such as Multilayer Perceptron (MLP), Support Vector Machine (SVM), Random Forest (RF), Convolutional Neural Networks (CNN), and Recurrent Neural Network (RNN), have been employed in PBR prediction.

Generally speaking, sequence-based methods rely solely on protein sequences for feature extraction. One of the concerns of these approaches is introducing or designing more useful features. For example, Li et al. [24] utilize three novel sequence-based features: HSP, position information and ProtVect. Stringer et al. [25] combine traditional sequence-based features with predicted structure features. Hosseini et al. [26] mainly apply multiple sequence alignment features with attention to create a 768-dimensional vector for each amino acid residue. Those features are generated by various bioinformatics tools which cost much extra time. Moreover, those sequence-based methods are very unstable and poorly generalized [27] as unbalanced benchmark datasets are used for PBRs prediction. The adoption of the ensemble learning method is a popular way to overcome this limitation.

In contrast, structure-based methods typically incorporate both sequence and structural data. These approaches focus on taking full advantage of protein conformation information based on graph representation learning and end-to-end deep learning models. Fout et al. [28] introduced Graph Convolutional Network (GCN) for PPIs prediction. Recently, GCN and its varieties are widely used [29,30]. The end-to-end deep learning models include geometric deep learning [31] and 3D CNN models [32]. The difference between graph representation learning and end-to-end deep learning models is that the graph-based methods need to take protein embeddings as input, but the end-to-end deep learning methods don’t.

As evident in Table 1, MSA (Multiple Sequence Alignment), predicted structural, and protein language model (PLM) features are commonly employed for protein representation learning. MSAs, derived from tools like PSI-BLAST [33], HHblits [34], or MMseqs2 [35], capture evolutionary information. Predicted structural information, such as secondary structure and solvent accessibility, derived from tools like NetSurfP-3.0 [36] or PSIPRED [37], could significantly enhance representation richness. Additionally, physicochemical properties and hydrophobicity are often considered. Most PBRs prediction methods concatenate these features directly into a matrix for embedding each input protein and then feed them into machine learning or deep learning models for predicting. Although this operation is simple, it consumes extra time and introduces uncertainty in feature selection. Inspired by natural language process technique and recent advancements in protein language models (PLMs), exemplified by ProtTrans [38] and ESM [39], have revolutionized protein representation. PLMs have shown utility in diverse biological tasks, including epitope prediction [40], protein-metal ion binding site prediction [41], and ligand binding site prediction [42]. Their integration into PBRs prediction models is increasingly essential.

The predominant approach for aggregating target and neighboring residue features is the sliding window strategy. This involves scanning a protein sequence with a fixed-size window and concatenating residue features within the window. While effective, the optimal window size varies across PBRs types and datasets. Previous studies have explored different window sizes, with 9 being common for protein-protein interactions [43,44]. However, determining the optimal window size is time-consuming and its impact on performance is not always consistent. Furthermore, the influence of neighboring residues on the central residue’s interaction potential is complex and not captured effectively by fixed-size windows. Consequently, there is a compelling need for more sophisticated methods to capture diverse residue dependencies without relying on arbitrary window sizes.
biomolecules-14-01220-t001_Table 1Table 1Summary of recent studies that focus on sequence-based PBRs prediction of different types.PBRs TypeMethodModel TypeMSAFeaturePredictedStructural FeaturePLMFeatureWindowSizePSPPIDER [45]MLP✓××11ISIS [46]MLP✓✓×-PSIVER [47]Naïve Bayes✓✓×9LORIS [48]LR✓✓×9SPRINGS [49]MLP✓✓×9CRFPPI [50]RF✓✓×9SPRINT [51]SVM✓✓×9SSWRF [52]SVM and RF✓✓×9RFPPI [53]RF×✓×9SeRenDIP [54]RF✓✓×9DLPred [55]LSTM✓××-Lu et al. [43]CNN and Attention✓✓×5N+S+PSCRIBER [56]LR✓✓×11ProNA2020 [57]MLP✓✓✓11PROBselect [58]LR and SVM✓✓×-DELPHI [24]CNN and GRU✓✓✓31PPISP-XGBoost [59]XGBoost✓✓×9PIPENN [25]CNN and RNN✓✓×9AttCNNPPI [44]CNN and Attention✓✓×5PITHIA [26]CNN and RNN✓✓×9StackingPPInet [60]DCNN and Attention✓××32EDLMPPI [27]GRU××✓25D-PPIsite [61]CNN✓✓×17ISPRED-SEQ [62]CNN✓×✓31EnsePPIs [63]CNN××✓-DeeBSRPred [64]MLP✓✓×11DeepRObind [65]CNN and Attention ✓ ✓×25PaproABC [66]RF×××-Parapred [67]CNN and RNN×××-Ag-Fast-Parapred [68]CNN and RNN×××-proABC-2 [69]CNN✓××-DeepANIS [70]BiLSTM and Transformer✓✓×-Lu et al. [71]CNN and BiLSM ✓ ✓×-EpBepipred-1.0 [72]Hidden Markov×××11AAPPred [73]MLP×××7CBTOPE [74]SVM×××13Bepipred-2.0 [75]RF✓✓×9SeRenDIP-CE [76]RF✓✓×9SEMA [77]MLP×× ✓9**Notes**: **P** stands for methods are evaluated on **P**rotein-protein binding residues. **N+S+P** indicates methods are evaluated on datasets containing protein-**N**ucleotide (DNA, RNA) binding residues, protein-**S**mall ligand binding residues and protein-**P**rotein binding residues. **Pa** and **Ep** are short for paratope and epitope which are binding residues from antibody and antigen interaction interface, respectively. **MSA feature** means protein feature returned by running multiple sequence alignment tools such as PSI-BLAST, HHsuits and MMseqs2. **PLM feature** means protein feature generated by protein language models such as ESM and ProtTrans. ✓ means this feature is used and × means this feature is not used.

In this paper, we design a novel sequence-based PBRs prediction framework PMSFF. The framework of PMSFF is presented in Figure 1. PMSFF improves the traditional sliding window approach and utilizes information on more scales through multi-scale feature fusion. PMSFF could handle multiple PBRs, provide better prediction results and spend less running time compared with the current prediction approaches. The contributions of our work are summarized as follows:

We construct multi-scale features through multi-size windows for applying more key neighboring residues and multi-size kernels on various scales. Inspired by the NLP technique, we take a protein sequence as a sentence and utilize bidirectional GRU for multi-scale feature fusion. This allows for better matching of different types of PBRs and obtaining proper residue local context and global semantic information.To leverage transfer learning, PMSFF incorporates ProtT5-XL-UniRef50 (ProtT5), a variant of the ProtTrans model trained on the UniRef50 dataset, for protein feature generation. A comparative analysis between ProtT5 features and traditional concatenated features reveals a substantial reduction in computational time while achieving significantly improved prediction performance.We collect and reorganize benchmark PBRs datasets covering most of the categories, including PBRs from protein interactions with nucleotide (DNA, RNA), small ligand, common protein, heteromeric protein, homomeric protein as well as antibody and antigen interaction. The results on these datasets demonstrate the superior performance of PMSFF compared with the state-of-the-art methods.

## 2. Materials and Methods

### 2.1. Benchmark Dataset Construction

To evaluate the performance of PMSFF on different types of PBRs and make a fair comparison with the state-of-the-art methods, we collect various datasets from previous studies and utilize the same way for training and evaluation [25,54,67].

Table 2 shows the summary of all datasets used in this study. Specifically, the NSP6373 and NSP448 datasets contain multiple PBRs from protein interactions with nucleotide (DNA, RNA), small ligands and other types of protein. Other datasets consist of PBRs from type-specific interactions between protein-protein, heterodimer, homodimer and antibody-antigen.

To train our model, we utilize the NSP6373 dataset, derived from the BioDL_A_TR dataset employed in PIPENN [25]. For evaluation, we employ the widely used NSP448 dataset [56] and its subset, NSP355, derived from DELPHI [24]. These datasets have been shown to exhibit low sequence similarity (<25%) [25]. To further mitigate sequence homology, we applied ECOD with an E-value cutoff of 0.001 [78,79], resulting in a non-redundant NSP6373 training set of 6373 protein sequences. It’s important to note that NSP6373, NSP448, and NSP355 encompass PBRs involved in interactions with nucleotides, small ligands, and other proteins. We use the “NSP” prefix to denote this broad PBRs type. Residues are defined as binding if an atom is within 0.5 Å plus the Van der Waals radii of any partner protein atom [25]. While the NSP6373 and NSP448 datasets encompass a wide range of protein-binding residues (PBRs) [56], it’s important to consider that PBRs from specific protein interactions can exhibit unique characteristics. To assess model performance on these specialized PBRs, we incorporate additional datasets focused on particular protein interfaces.

The P346 and P70 datasets are derived from Dset186, Dset72 [47], and PDBset164 [48], which were originally used to train DeepPPISP [80]. Following DeepPPISP’s methodology, we constructed a non-redundant dataset with a sequence identity threshold of 25% and split it into training (346 proteins) and test (72 proteins) sets. This process ensures minimal sequence homology between the two sets. Both P346 and P70 contain PBRs from common heterodimeric interfaces.

The N376, N38, S3874, and S354 datasets are derived from the BioDL_S_TR, ZK448_S_TE, BioDL_N_TR, and ZK448_N_TE datasets, respectively, which were employed for training and testing PIPENN [25]. To ensure dataset independence, homologous proteins between training and test sets were removed using ECOD. The resulting datasets contain type-specific PBRs involved in interactions with nucleotides (DNA, RNA) and small ligands.

The HHC489 dataset, originally used for training RFPPI [53], comprises PBRs from heterodimer and homodimer interfaces. RFPPI constructed a homodimer dataset (HM_479) with a 25% sequence identity threshold and derived a test set (Ho95) by randomly selecting 20% of HM_479. Similarly, a heterodimer dataset (Dset_119) was trained and tested on Dset_48 (renamed He48) with a 25% sequence identity cutoff. To create a balanced training set, we combined HM_479 and Dset_119, removing any proteins homologous to Ho95 or He48, resulting in the HHC489 dataset containing 489 proteins. Consistent with RFPPI [53], residues in HHC489, Ho95, and He48 are defined as binding if both their pre-association accessible surface area (ASA) and post-association buried surface area (BSA) exceed 6.0 Å.

Antibody-antigen interactions represent a specific type of protein interface [81]. Paratopes and epitopes are the corresponding binding residues on antibodies and antigens, respectively. To evaluate paratope and epitope prediction methods, the Pa277 and Ep280 datasets are commonly employed [67,76]. Liberis et al. curated the Pa277 dataset comprising 277 complexes from the SAbDab database [82], defining paratopes as antibody residues within 4.5 Å of any antigen atom. Similarly, Hou et al. constructed the Ep280 dataset containing 280 complexes from SAbDab, designating residues within 6.0 Å of the antibody as epitopes [76].

### 2.2. Framework Architecture of the PMSFF

As illustrated in Figure 1A, our proposed PMSFF model comprises three core components: an attention layer, a convolutional neural network (CNN) layer, and a gated recurrent unit (GRU) layer. The model processes an input matrix of residue feature vectors. The attention layer captures contextual information from neighboring residues using a refined sliding window approach across multiple scales. Subsequently, the CNN layer extracts local features through one-dimensional convolutions with varying kernel sizes, followed by max pooling. The bidirectional GRU layer models long-range dependencies among residues. Finally, a fully connected layer predicts the binding probability for each residue.

#### 2.2.1. The Input Representation of Proteins

As depicted in Figure 1, we employ the ProtT5 protein language model for feature extraction. ProtT5 is pre-trained on the BFD dataset [83] and subsequently fine-tuned on UniRef50 [84]. The resulting 1024-dimensional residue embeddings from the final ProtT5 layer serve as our primary protein representation. To assess the impact of different feature encoding strategies, we compare this approach with a standard concatenation of traditional residue features.

As indicated in Table 1, MSA and predicted structural features are prevalent in PBRs prediction. Our feature set, consistent with previous work [44], comprises one-hot encoded residue types, position-specific scoring matrices (PSSMs) generated by MMseqs2 [35], and predicted structural features including secondary structure, solvent accessibility, and backbone dihedral angles from NetSurfP-2.0 (NSP2) [85] or NetSurfP-3.0 (NSP3) [36]. While NSP2 relies on MMseqs2 for feature extraction, NSP3 incorporates the ESM language model [39]. This results in a 52-dimensional residue representation, aligning with our prior work [43]. Given ProtT5’s demonstrated efficacy in residue-level classification [41], we prioritize its use for feature generation. ProtT5 produces a set of embedding vectors for each protein, forming the input matrix for our PMSFF model, which is formulated as:(1)$$S={\left[r_{1},r_{2},r_{3},\cdots ,r_{i},\cdots ,r_{l}\right]}^{T},r_{i}\in R^{d}$$
where *l* is the protein length, *d* equals to the dimension of the residue feature vector and $r_{i}$ is the feature vector of *i*-th residue.

#### 2.2.2. Attention Layer

While the sliding window approach effectively captures local residue dependencies [53], its performance is hindered by the complex and varied nature of residue interactions. Optimal window sizes can differ across PBRs types and even within the same PBRs category, as evidenced in Table 1. Our previous work demonstrated the efficacy of attention mechanisms in aggregating target and neighboring residues [43], with optimal performance achieved at a window size of 5. To accommodate the diverse characteristics of PBRs, we introduce a multi-scale attention mechanism that employs multiple window sizes.

As depicted in Figure 1A, our attention layer assigns weights to neighbor residues within a specified window, generating a context vector through weighted averaging. Mathematically, the neighboring residue $r_{j}$ of the target residue $r_{i}$ within the window is defined as:(2)$$r_{j}\in \left\{r_{i-w},r_{i-w+1},\ldots ,r_{i},\ldots ,r_{i+w-1},r_{i+w}\right\}$$

To capture complex patterns between target residue and neighboring residues, we improve the sliding window approach by utilizing *n* windows of different sizes. The neighboring residue $r_{j}^{n}$ of target residue $r_{i}$ in the *n*-th window is shown as follow:(3)$$r_{j}^{n}\in \left\{r_{i-w_{n}},r_{i-w_{n}+1},\ldots ,r_{i},\ldots ,r_{i},r_{i+w_{n}-1},r_{i+w_{n}}\right\}$$

As our previous work [44], we utilize an attention mechanism to enable the target residue learning to pay different attention to each neighboring residue. The typical additive model is used in this study. In the *n*-th window, the similarity score $score(r_{i},r_{j}^{n})$ between target residue $r_{i}$ and neighboring residue $r_{j}^{n}$ and the attention weight $\alpha_{i,j}^{n}$ are calculated as:(4)$$score\left(r_{i},r_{j}^{n}\right)=v_{a}^{n}\tanh\left(W_{a}^{n}\left[r_{i}⊕r_{j}^{n}\right]\right)$$
(5)$$\alpha_{i,j}^{n}=\frac{exp(score\left(r_{i},r_{j}^{n}\right))}{\sum_{j^{\prime }}exp(\left(score\left(r_{i},r_{j}^{n}\right)\right)}$$

The residue pair correlation score $score(r_{i},r_{j})$ is computed by a two-layer neural network described in formula (4) in which $v_{a}^{n}$ and $W_{a}^{n}$ are weight matrixes when learning within *n*-th windows. And neighboring residues $r_{j,j\neq i}$ with larger scores contribute more to the context vector $g_{i}$. The attention weights $\alpha_{i,j}$ should satisfy the following restrictions: $\alpha_{i,j}\geq 0$ and $\sum_{i}\alpha_{i,j}=1$.

The context vector $g_{i}$ of the *n*-th window is constructed by scoring and combing neighboring residue vectors in a weighted sum:(6)$$g_{i}^{n}=\sum_{j\neq i}\alpha_{i,j}^{n}*r_{j}^{n}$$

The output of the attention layer is the concatenation of the original residue vector and context vectors learning from sliding windows of different sizes shown as follows:(7)$$r_{i}^{\prime }=r_{i}⊕g_{i}^{1}⊕\cdots ⊕g_{i}^{n}$$

#### 2.2.3. CNN Layer

Convolutional neural networks (CNN) have been used in various bioinformatics tasks such as protein structure prediction [86] and protein-compound affinity prediction [87]. The usage of multi-size kernels in CNN is helpful for extracting information because it can account for various scales which has improved the performance in text classification [88] and image classification [89]. Inspired by TextCNN [88], we utilize multi-size kernels in the CNN layer of PMSFF. This architecture allows the network to extract information over multiple scales. The input of the CNN layer is the output of the attention layer which can be shown as follows:(8)$$S^{\prime }={\left[r_{1}^{\prime },r_{2}^{\prime },r_{3}^{\prime },\cdots ,r_{i},\cdots ,r_{l-1},r_{l}^{\prime }\right]}^{T},r_{i}\in R^{4d}$$

And, *m* kernels of different sizes are used in this study when carrying out our experiments. Therefore, every convolutional operation is shown as follows:(9)$$c_{i}^{m}=f_{c}^{m}\left(W_{c}^{m}*r_{i}^{\prime }+b_{c}^{m}\right)$$
where $f_{c}^{m}$ is *m*-th non-linear activation function, and $r_{i}^{\prime }$ is the output of the CNN layer.

As illustrated in Figure 1C, the output of the CNN layer is a feature matrix of dimensions *l* × 3*d*, where *l* is the sequence length and *d* is the number of output channels per convolutional filter. A subsequent max-pooling layer with a filter size of 5 and zero padding downsamples the feature matrix, generating the feature vector $p_{i}$ as input to the GRU layer.
(10)$$p_{i}=maxpooling\left(c_{i}^{1}⊕\cdots ⊕c_{i}^{m}\right)$$

#### 2.2.4. GRU Layer

Recurrent Neural Networks (RNN) excel at processing sequential data [90]. While both Long Short-Term Memory (LSTM) [91] and Gated Recurrent Unit (GRU) [92] are RNN variants, GRU’s computational efficiency makes it our preferred choice. To capture intricate residue interactions within the protein sequence, we incorporate a GRU layer in PMSFF. The input to this GRU layer is a matrix of residue feature vectors, formed by concatenating the outputs of all CNN channels. The resulting feature vector for residue *i* is denoted as $p_{i}$:(11)$$p_{i}=\left[e_{1},e_{2},e_{3},\cdots ,e_{3k}\right]$$

To capture both forward and backward contextual information within protein sequences, analogous to natural language processing, we employ a bidirectional gated recurrent unit (GRU) with a hidden state size of 64. The GRU calculations are as follows:(12)$$z_{t}=\sigma \left(W_{z}\cdot \left[h_{t-1},e_{t}\right]+b_{z}\right)$$
(13)$$r_{t}=\sigma \left(W_{r}\cdot \left[h_{t-1},e_{t}\right]+b_{r}\right)$$
(14)$$\tilde{h_{t}}=tanh\left(W_{c}\cdot \left[r_{t}*h_{t-1},e_{t}\right]+b_{c}\right)$$
(15)$$h_{t}=\left(1-z_{t}\right)*h_{t-1}+z_{t}*\tilde{h_{t}}$$

There are two gates consisting of one reset gate $z_{t}$ with corresponding weight matrix $W_{z}$, and one bias $b_{z}$; one update gate $r_{t}$ with corresponding weight matrix $W_{r}$, and a bias $b_{r}$. And, tanh is the element-wise hyperbolic tangent, σ is the logistic sigmoid function, $W_{c}$ is weight matrix, $e_{t}$ and $h_{t-1}$ are inputs, and $h_{t}$ is output.

Bidirectional GRU can learn information of input sequence from forward and backward. As shown in Figure 1A, the GRU layer in PMSFF contains two sub-networks for the left and right sequence contexts. For the *i*-th residue in the input antigen sequence, we combine the forward pass output $\vec{h_{i}}$ and backward pass output $\overset{\leftarrow }{h_{i}}$ by concatenating them:(16)$$h_{t}^{\prime }=\left[\vec{h_{t}}⊕\overset{\leftarrow }{h_{t}}\right]$$

As shown in Figure 1C, the output $h_{t}^{\prime }$ is fed to a fully connected layer. The calculation of binding probability $o_{i}$ for each input residue is shown as:(17)$$o_{i}=f_{o}\left(W_{o}h_{t}^{\prime }+b_{o}\right)$$

## 3. Results

In this section, we first investigate the effects of multi-scale and ProtT5 features on the PMSFF model. Then we compare the proposed model with the SOTA approaches on different evaluation datasets. The evaluation metrics and the implementation details have been displayed in Section A.1 and Section A.2 respectively.

### 3.1. The Effectiveness of Multi-Scale Feature

To enhance PBRs prediction performance, we incorporate multi-scale features by applying multiple window sizes within the attention layer and kernel sizes within the CNN layer. A GRU layer is further employed to capture long-range dependencies. Our PMSFF model utilizes three window sizes (5, 15, 25), three kernel sizes (3, 5, 7), and a single GRU layer.

To assess the impact of these components, we conduct experiments varying the number of windows, kernels, and GRU layers from 0 to 4. A value of 0 indicates the absence of the corresponding layer. Each model configuration is trained and tested five times on the NSP6373 and NSP448 datasets for robust evaluation. Mean performance metrics and standard errors are presented in Table 3.

As shown in Table 3, our proposed PMSFF model consistently outperforms other configurations across all evaluation metrics. Increasing the number of attention windows generally improves performance, but excessive windows lead to overfitting. While a CNN layer without kernels yielded reasonable results, employing three kernels was optimal for capturing local features. The GRU layer demonstrated the best performance with a single layer, as additional layers tended to overfit. Overall, these findings suggest that while multi-scale features can enhance model performance, an appropriate balance between complexity and model capacity is essential.

Building upon our previous work [43], we initially selected a window size of 5 for capturing local residue interactions. To incorporate longer-range dependencies, four window sizes of 15 and 35 were introduced. Our results indicate that employing three window sizes (5, 15, 25) outperforms configurations with fewer windows, suggesting the benefit of capturing information from diverse neighborhood scales. However, excessively large window sizes, as demonstrated by the W4K3GRU1 model, can introduce noise and hinder performance.

Inspired by TextCNN [88] and DeepPPISP [80], we incorporate 1D convolutional layers into PMSFF to extract deeper residue-level features. Given that the attention layer already captures local dependencies, we focus on capturing more complex patterns within the sequence. We employ kernel sizes of 3, 5, and 7, consistent with our previous work [43]. Results in Table 3 indicate that three kernels optimize performance while increasing the number of kernels to four (sizes 3, 5, 7, 9) degrades accuracy, similar to the observations made with multiple windows. This suggests that excessive feature engineering can introduce noise and hinder model performance.

To effectively fuse multi-scale features and capture intricate long-range dependencies within the protein sequence, we incorporate a GRU layer into PMSFF. Table 3 demonstrates that a single-layer GRU outperforms deeper GRU architectures, aligning with our previous findings in structure-based epitope prediction [93]. The combination of CNN and GRU layers, integrating spatial and temporal information processing, has been shown to enhance performance in various classification tasks [89]. This synergistic interaction between convolutional and recurrent layers is crucial for learning complex protein representations.

In summary, our proposed PMSFF model integrates attention, convolutional, and recurrent layers to effectively capture diverse feature representations for PBRs prediction. The model employs three windows in the attention layer, three kernels in the CNN layer, and a single-layer GRU to optimize performance. Each component contributes significantly to the model’s overall predictive power.

### 3.2. The Evaluation Results of ProtT5 Feature

Traditional PBRs prediction methods often rely on concatenating multiple residue features, including MSA and predicted structural information. This approach is computationally demanding and requires extensive experimentation to determine the optimal feature set. To address these limitations, we propose leveraging ProtT5-generated embeddings as a comprehensive protein representation. To evaluate the effectiveness of ProtT5, we compare PMSFF’s performance using ProtT5 embeddings against models employing concatenated features comprising PSSMs from MMseqs2 and structural information from either NSP2 or NSP3. We conduct five-fold cross-validation on the NSP6373 and NSP448 datasets to assess model robustness.

As illustrated in Figure 2, the computational efficiency of generating concatenated features, specifically those incorporating NSP2 [85] or NSP3 [36] predictions, is significantly higher than that of using ProtT5 embeddings. For protein sequences ranging from 200 to 700 residues, NSP2 and NSP3 require approximately 10 s and 3 s, respectively, while ProtT5 completes the process in less than 0.1 s. This substantial time savings enables the application of ProtT5 [38] to larger datasets, potentially enhancing model performance and prediction speed.

ProtT5 embeddings consistently outperform concatenated feature sets, demonstrating the effectiveness of transfer learning. As illustrated in Figure 2B, ProtT5 achieves superior performance across all evaluation metrics, with statistically significant improvements. The ROC and PR curves in Figure 2C,D further emphasize this advantage. Compared to NSP2+MMseqs2 features, ProtT5 exhibits absolute improvements of 0.157 in AUROC and 0.207 in AUPRC. When compared to NSP3+MMseqs2 features, these improvements are 0.132 and 0.187, respectively.

### 3.3. Performance Comparison with Other Baselines on NSP448 Dataset

To evaluate the performance of PMSFF, we compare it against 19 published sequence-based PBRs prediction methods on the NSP448 dataset. Table 4 presents the dataset-level performance metrics for all compared methods, calculated across all residues in NSP448. Results were obtained from various sources, including the original publications, the SCRIBER study [56], and web server predictions.

As shown in Table 4, PMSFF surpasses all compared methods in terms of F1-score (0.694) and AUPRC (0.514). These metrics are particularly sensitive to class imbalance, making PMSFF well-suited for PBRs prediction. Compared to the second-best performing method, D-PPIsite [61], PMSFF exhibits substantial improvements of 0.038 and 0.214 in the F1-score and AUPRC, respectively. While PMSFF’s MCC (0.388) is slightly lower than the top-performing D-PPIsite (0.399), the overall results highlight the effectiveness of our proposed method. To further illustrate the strengths of PMSFF, we provide the visual prediction results of a representative protein in the Appendix A. As illustrated by the example shown in Appendix A, PMSFF accurately predicts a greater portion of a typical PBR than 13 other commonly used methods.

To expand the comparative analysis, we constructed the NSP355 dataset, a subset of NSP448 [24]. Performance metrics for additional methods, including DELPHI [24] and results obtained from PIPENN, DeepBSRPred, and DeepRObind web servers, are presented in Table 5. Note that DeepBSRPred’s binary output format precluded the calculation of F1 and MCC metrics. PMSFF demonstrates superior performance in terms of F1-score (0.499) and AUPRC (0.502) on the NSP355 dataset. While PMSFF’s MCC value of 0.389 closely trails D-PPIsite’s top performance of 0.390, the overall results underscore the effectiveness of our proposed method.

Leveraging results available on the DELPHI website (https://delphi.csd.uwo.ca/ (accessed on 26 June 2023)), we calculated AUROC and AUPRC for each protein sequence in the NSP355 dataset across PMSFF and 11 competing methods. Figure 3 depicts these results, highlighting PMSFF’s superior performance with median AUROC (0.833) and median AUPRC (0.567). PIPENN emerges as the second-best performer, achieving median AUROC and AUPRC values of 0.722 and 0.372, respectively. Notably, PMSFF surpasses PIPENN by a substantial margin, demonstrating improvements of 0.111 in median AUROC and 0.195 in median AUPRC.

### 3.4. Evaluation Results on Type-Specific Datasets

As Table 6 shows, we compare PMSFF with other methods on type-specific PBRs from protein-protein interaction, protein-small ligand interaction, protein-nucleotide interaction, heterodimer interface, homodimer interface and antibody-antigen interaction.

To evaluate PMSFF’s performance on common protein-protein interactions, we trained the model on the P346 dataset and tested it on the P70 dataset, following the protocol of DeepPPISP [80]. Table 6 presents the results of PMSFF and other competing methods, including those reported in DeepPPISP [80] and EnsemPPIS [63]. PMSFF demonstrates superior performance across most metrics, with particularly strong results in AUPRC (0.435), surpassing the second-best performers (EGRET and EnsemPPIS) by 0.03. Notably, DeepPPISP and EGRET are structure-based methods, while PMSFF is solely sequence-based.

To assess PMSFF’s performance on specific PBRs types, we re-trained and evaluated the model on datasets representing interactions with small ligands (S), nucleotides (N), heterodimers (He), homodimers (Ho), paratopes (Pa), and epitopes (Ep). Figure 4 presents a comparative analysis of AUROC values for PMSFF and other methods on the S354, N38, He48, and Ho95 datasets.

As illustrated in Figure 4, PMSFF consistently demonstrates superior AUROC performance across most PBRs types. However, for protein-small ligand binding residues, PIPENN [25] marginally outperforms PMSFF. It is noteworthy that while bindEmbed21(DL) [42] also leverages ProtT5 embeddings, its performance lags behind both PIPENN and PMSFF in this specific task. PMSFF obtains better AUROC than PIPENN [25] when predicting PBRs from protein-nucleotide interaction, heterodimer interface and homodimer interface. It indicates the advantage of PMSFF for predicting type-specific PBRs.

To evaluate paratope prediction performance, PMSFF was subjected to 10-fold cross-validation on the Pa277 dataset. Figure 4 clearly demonstrates PMSFF’s superiority over other paratope prediction methods. Similarly, for epitope prediction, 10-fold cross-validation was performed on the Ep280 dataset, with PMSFF achieving the highest AUROC value. It is noteworthy that Beipred-3.0 [40], a recently proposed method leveraging the ESM language model [39] for antigen representation, serves as a strong baseline.

## 4. Conclusions

Recent advancements in sequence-based PBRs prediction have encountered challenges in determining optimal feature scales and overcoming the limitations of the sliding window approach. To address these issues, we propose PMSFF, a generalizable framework for multi-PBR prediction. By leveraging the ProtT5 language model [38] for feature extraction, PMSFF efficiently generates comprehensive protein representations without relying on external tools or time-consuming feature engineering.

PMSFF incorporates multi-scale features and employs attention, convolutional, and recurrent layers to capture complex residue interactions. Extensive experiments on diverse benchmark datasets, encompassing various PBRs types, demonstrate PMSFF’s superior performance compared to existing methods.

Although PMSFF achieves the best performances on multiple PBRs predictions, it has a limitation in that partner information isn’t used. There are some proposed partner-specific PBRs prediction methods predicting binding residues for a particular pair interaction [94]. We will design a novel deep learning model applying for partner-specific multiple PBRs prediction in the future.

## Figures and Tables

**Figure 1 biomolecules-14-01220-f001:**
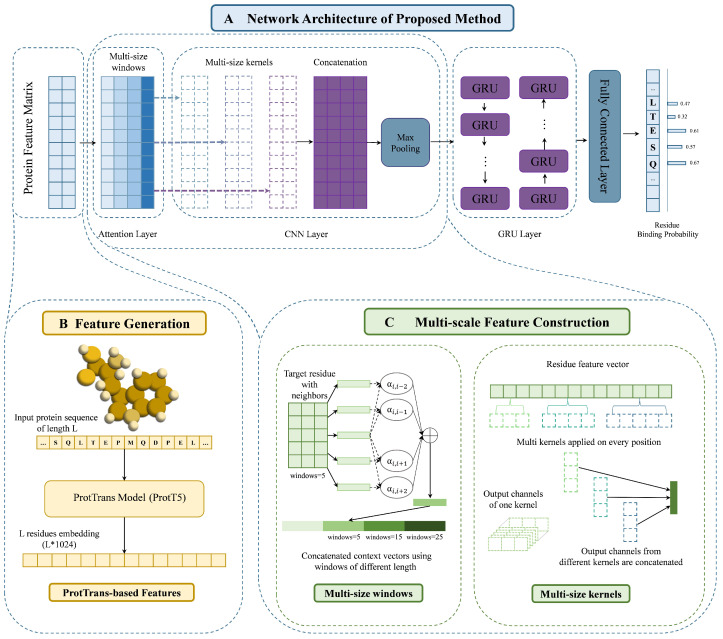
The framework of PMSFF. (**A**) Protein embedding is obtained by the ProtT5 model. (**B**) The construction of multi-scale features. **Left**: Context vector is generated by attention mechanism using sliding window approach. Multi-size windows are used for adapting different types of PBRs and capturing complex patterns between target residue and neighboring residues. **Right**: Multi-size kernels are utilized in convolutional neural networks on input residue feature vectors for information on more scales. The output channels of each kernel are concatenated. (**C**) Details of the framework architecture. Our proposed framework PMSFF mainly consists of three parts: the attention layer, CNN layer and GRU layer.

**Figure 2 biomolecules-14-01220-f002:**
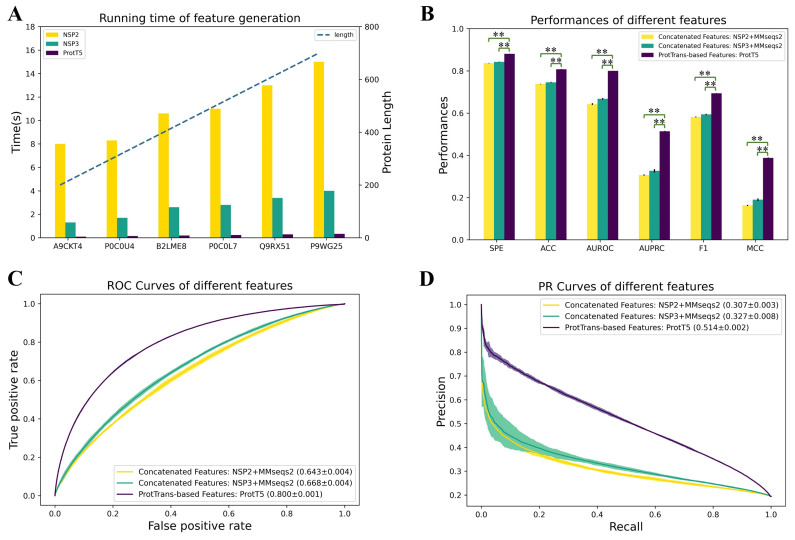
Comparison between concatenated features and ProtT5 features on NSP448 set. (**A**) The time of running NetSurfP-2.0 (NSP2) [85], NetSurfP-3.0 (NSP3) [36] and ProtT5 [38] on various lengths of protein sequences. (**B**) Prediction performance comparison on SPE, ACC, AUROC, AUPRC, F1 and MCC. We train and test PMSFF using three kinds of features five times. ** denote that concatenated features are significantly worse than ProtT5 features with *p* < 0.005. (**C**) The comparison of ROC curves of three kinds of features. (**D**) The comparison of PR curves of three kinds of features.

**Figure 3 biomolecules-14-01220-f003:**
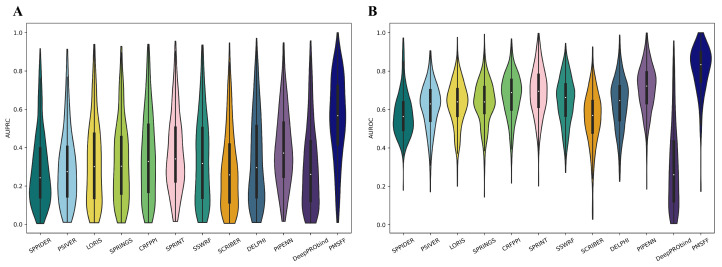
Protein-level performance comparison of PMSFF and other methods on NSP355. (**A**) The distributions of per-protein AUROC values where the thick vertical lines represent the first quartile, median (white dot) and third quartile, whiskers denote the minimal and maximal values. (**B**) The distributions of per-protein AUPRC values as AUROC values.

**Figure 4 biomolecules-14-01220-f004:**
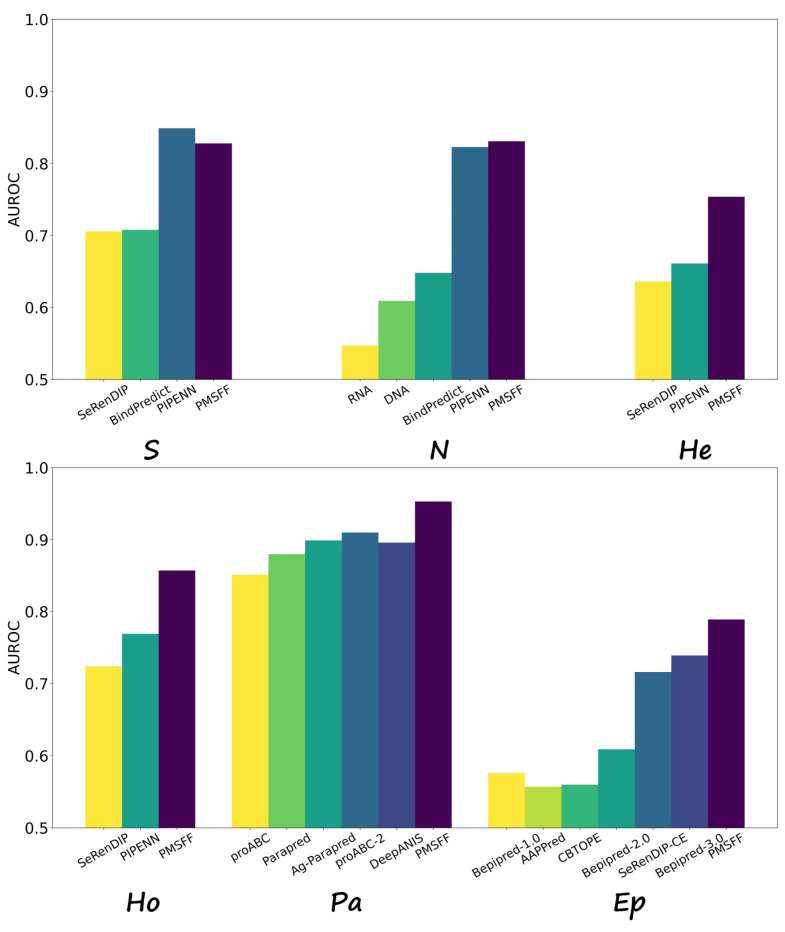
Comparison of PMSFF and other methods on type-specific PBRs. **N** and **S** mean the protein-Nucleotide (DNA, RNA) binding residues and protein-Small ligand binding residues. **He** and **Ho** stand for the binding residues from heterodimers and homodimers. **Pa** and **Ep** are short for paratope and epitope which are binding residues from antibody and antigen interaction interface, respectively.

**Table 2 biomolecules-14-01220-t002:** Summary of experimental datasets.

Dataset	Proteins	Residues	Binding Residues	Evaluation Method	PBRs Type
NSP6373	6373	2,068,793	203,880 (10.60%)	Hold-out (Training Set)	N+S+P
NSP355	355	95,940	11,467 (11.95%)	Hold-out (Test Set)	
NSP448	448	116,500	22,643 (19.44%)	Hold-out (Test Set)	
P346	346	75,204	11,027 (14.66%)	Hold-out (Training Set)	P
P70	70	11,791	2330 (19.76%)	Hold-out (Test Set)	
N376	376	124,476	15,821 (12.71%)	Hold-out (Training Set)	S
N38	38	10,212	1233 (12.07%)	Hold-out (Test Set)	
S3874	3874	1,246,682	129,245 (10.37%)	Hold-out (Training Set)	N
S354	354	99,318	7175 (3.20%)	Hold-out (Test Set)	
HHC489	489	126,997	25,564 (20.13%)	Hold-out (Training Set)	He+Ho
He48	48	14,056	1313 (9.34%)	Hold-out (Test Set)	
Ho95	95	24,766	5564 (22.47%)	Hold-out (Test Set)	
Pa277	277	115,341	6151 (5.33%)	K-fold Cross-validation	Pa
Ep280	280	75,956	6971 (9.18%)	K-fold Cross-validation	Ep

**Notes**: **N+S+P** indicates methods are evaluated on datasets containing protein-nucleotide (DNA, RNA) binding residues, protein-small ligand binding residues and protein-protein binding residues. **P** stands for methods that are tested on protein-protein binding residues. **He** and **Ho** are short for heterodimer and homodimer, respectively. **HHC** means the dataset contains binding residues from heterodimer and homodimer. **Pa** and **Ep** are short for paratope and epitope which are binding residues from antibody and antigen interaction interface, respectively. **Hold-out** and **K-fold Cross-validation** are two type of ways for evaluating model performance. For Hold-out validation, all data is split into three parts: training, validation and testing. For K-fold Cross-validation, all data is split into K subsets and the model is trained on K-1 subsets and tested on the last subset K times. To make a fair comparison, we utilize the same datasets and evaluating methods as competing models.

**Table 3 biomolecules-14-01220-t003:** Impact of multi-scale features on the NSP448 dataset.

Methods		SPE	ACC	F1	MCC	AUROC	AUPRC
Attention	W0K3GRU1	0.871(0.001)	0.792(0.002)	0.669(0.004)	0.337(0.008)	0.764(0.005)	0.469(0.009)
	W1K3GRU1	0.873(0.001)	0.796(0.001)	0.674(0.001)	0.347(0.003)	0.769(0.002)	0.481(0.004)
	W2K3GRU1	0.874(0.002)	0.797(0.003)	0.676(0.006)	0.351(0.011)	0.772(0.009)	0.483(0.009)
	W4K3GRU1	0.871(0.001)	0.793(0.001)	0.669(0.002)	0.339(0.005)	0.765(0.005)	0.472(0.002)
CNN	W3K0GRU1	0.874(0.000)	0.798(0.000)	0.677(0.001)	0.354(0.001)	0.769(0.001)	0.482(0.001)
	W3K1GRU1	0.869(0.002)	0.789(0.004)	0.663(0.006)	0.326(0.012)	0.759(0.008)	0.454(0.010)
	W3K2GRU1	0.871(0.002)	0.792(0.003)	0.669(0.004)	0.337(0.008)	0.764(0.006)	0.468(0.008)
	W3K4GRU1	0.871(0.001)	0.792(0.002)	0.669(0.003)	0.337(0.007)	0.768(0.009)	0.465(0.006)
GRU	W3K3GRU0	0.874(0.002)	0.797(0.003)	0.676(0.005)	0.351(0.009)	0.777(0.007)	0.486(0.008)
	W3K3GRU2	0.872(0.002)	0.794(0.003)	0.671(0.004)	0.341(0.009)	0.769(0.008)	0.464(0.008)
	W3K3GRU3	0.869(0.002)	0.789(0.003)	0.663(0.005)	0.326(0.009)	0.760(0.005)	0.453(0.010)
Proposed	W3K3GRU1 (PMSFF)	**0.881(0.000)**	**0.808(0.001)**	**0.694(0.001)**	**0.388(0.003)**	**0.800(0.001)**	**0.514(0.002)**

**Notes**: **Attention** means multi-scale feature vary in attention layer. **CNN** means multi-scale features vary in the CNN layer. **GRU** means multi-scale feature fusion varying in the GRU layer. **W** indicates window. **K** indicates kernel. The number after W, K and GRU means the number of windows, kernels and GRU layers. The last row is our proposed method PMSFF. The best results are given in bold.

**Table 4 biomolecules-14-01220-t004:** Dataset-level comparison with competing methods on NSP448 dataset.

Methods	SPE	ACC	F1	MCC	AUROC	AUPRC
SPPIDER ^*a*^	0.870	0.781	0.198	0.071	0.517	0.159
SPRINT ^*a*^	0.873	0.781	0.183	0.057	0.570	0.167
PSIVER ^*a*^	0.874	0.783	0.191	0.066	0.581	0.170
SPRINGS ^*a*^	0.882	0.796	0.229	0.111	0.625	0.201
LORIS ^*a*^	0.887	0.805	0.263	0.151	0.656	0.228
CRFPPI ^*a*^	0.887	0.805	0.266	0.154	0.681	0.238
SSWRF ^*a*^	0.891	0.811	0.287	0.178	0.687	0.256
SCRIBER ^*a*^	0.896	0.821	0.333	0.230	0.715	0.287
DELPHI ^*b*^	0.901	0.829	0.371	0.272	0.737	0.337
PIPENN ^*b*^	0.870	0.785	0.385	0.254	0.729	0.357
AttCNNPPI ^*b*^	0.831	0.737	0.587	0.174	0.634	0.316
PITHIA ^*b*^	0.907	0.840	0.408	0.315	0.778	0.387
StackingPPINet ^*b*^	NA	NA	0.387	0.129	0.593	0.406
EDLMPPI ^*b*^	0.922	0.858	0.464	0.383	0.820	0.460
D-PPIsite ^*b*^	0.919	**0.859**	0.480	**0.399**	**0.824**	0.476
ISPRED-SEQ ^*b*^	NA	NA	0.470	0.390	0.820	NA
EnsePPIs ^*b*^	NA	0.821	0.385	0.291	0.770	0.354
DeepBSRPred ^*b*^	0.630	NA	0.380	0.260	0.740	0.340
DeepRObind ^*c*^	**0.949**	0.802	0.272	0.206	0.666	0.354
PMSFF	0.881	0.808	**0.694**	0.388	0.800	**0.514**

**Notes**: ^*a*^ Results reported by SCRIBER [56]. ^*b*^ Results taken from original published studies. ^*c*^ Results obtained by utilizing the web server. The best results are given in bold.

**Table 5 biomolecules-14-01220-t005:** Dataset-level comparison with competing methods on NSP355 dataset.

Methods	SPE	ACC	F1	MCC	AUROC	AUPRC
SPPIDER ^*a*^	0.889	0.804	0.180	0.068	0.515	0.138
SPRINT ^*a*^	0.886	0.801	0.168	0.054	0.571	0.150
PSIVER ^*a*^	0.888	0.803	0.177	0.065	0.583	0.155
SPRINGS ^*a*^	0.892	0.811	0.211	0.103	0.608	0.178
LORIS ^*a*^	0.896	0.818	0.241	0.137	0.637	0.203
CRFPPI ^*a*^	0.897	0.819	0.246	0.143	0.662	0.214
SSWRF ^*a*^	0.901	0.825	0.268	0.168	0.667	0.228
SCRIBER ^*a*^	0.908	0.838	0.322	0.230	0.719	0.275
DLPred ^*a*^	0.906	0.835	0.308	0.214	0.724	0.272
DELPHI ^*a*^	0.914	0.848	0.364	0.278	0.746	0.326
PIPENN ^*c*^	0.863	0.863	0.375	0.237	0.710	0.351
PITHIA ^*b*^	0.916	0.852	0.381	0.297	0.762	0.344
D-PPIsite ^*b*^	0.927	**0.871**	0.460	0.387	**0.822**	0.448
ISPRED-SEQ ^*b*^	NA	NA	0.460	**0.390**	0.820	NA
EnsePPIs ^*b*^	NA	0.821	0.385	0.291	0.770	0.354
DeepBSRPred ^*c*^	0.557	0.533	0.244	−0.018	NA	NA
DeepRObind ^*c*^	**0.957**	0.815	0.248	0.196	0.662	0.331
PMSFF	0.890	0.819	**0.499**	0.389	0.805	**0.502**

**Notes**: ^*a*^ Results reported by DELPHI [24]. ^*b*^ Results taken from original published studies. ^*c*^ Results obtained by utilizing the web server. The best results are given in bold.

**Table 6 biomolecules-14-01220-t006:** Dataset-level comparison with competing methods on P70 dataset.

Methods	PRE	SEN	ACC	F1	MCC	AUROC	AUPRC
PSIVER ^*a*^	0.253	0.468	0.653	0.328	0.138	NA	0.250
SPPIDER ^*a*^	0.209	0.459	0.622	0.287	0.089	NA	0.230
SPPIDER ^*b*^	0.240	0.315	0.667	0.273	0.063	0.518	0.235
SPRINGS ^*a*^	0.248	0.598	0.631	0.350	0.181	NA	0.280
ISIS ^*a*^	0.211	0.362	0.622	0.267	0.097	NA	0.240
RFPPI ^*a*^	0.173	0.512	0.598	0.258	0.118	NA	0.210
IntPred ^*a*^	0.247	0.508	0.672	0.332	0.165	NA	NA
DeepPPISP ^*a*^	0.303	0.577	0.655	0.397	0.206	0.671	0.320
DLPred ^*b*^	0.325	0.577	0.680	0.416	0.235	0.697	0.380
ProNA2020 ^*b*^	0.297	0.229	**0.741**	0.258	0.106	NA	NA
SCRIBER ^*b*^	0.274	0.569	0.616	0.370	0.159	0.635	0.307
DELPHI ^*b*^	0.319	0.604	0.667	0.418	0.236	0.690	0.360
EGRET ^*b*^	0.358	0.561	0.715	0.438	0.270	0.719	0.405
EnsemPPIS ^*b*^	0.375	0.532	0.732	0.440	0.277	0.719	0.405
AttCNNPPI ^*c*^	0.587	0.635	0.536	**0.610**	0.216	0.690	0.338
PMSFF	**0.629**	**0.679**	0.717	0.304	**0.653**	**0.747**	**0.435**

**Notes**: ^*a*^ Results reported by DeepPPISP [80]. ^*b*^ reported by EnsemPPIS [63]. ^*c*^ Results taken from original published studies. The best results are given in bold.

## Data Availability

The source code and datasets of our proposed method PMSFF are available at https://github.com/biolushuai/PMSFF-for-multiple-PBRs-prediction (accessed on 23 September 2024).

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
