# Peer review of "PMSFF: Improved Protein Binding Residues Prediction through Multi-Scale Sequence-Based Feature Fusion Strategy"

_biomolecules, 2024, doi:10.3390/biom14101220_

Round 1

Reviewer 1 Report

Comments and Suggestions for Authors

Summary of authors work:

The authors propose a novel methodology for predicting protein binding residues (PBR) that is called Multi-scale Sequence-based Feature Fusion (PMSFF). Their new strategy employs a pre-trained existing language model (ProtT5) for feature extraction combined with attention, convolutional, and recurrent layers. Their algorithm allows for multiple sliding windows and kernel sizes to be tested during the analysis. The authors compared their new framework to nineteen pre-existing methods (e.g., SPPIDER, DeepBSRPred, PIPENN) and used a variety of benchmark datasets that incorporated type-specific interactions such as protein-protein, heterodimer, homodimer, and antibody-antigen (e.g., NSP355, NSP448 from DELPHI). The authors found that ProtT5 was computationally more efficient for generating concatenated features compared to NSP2 and NSP3 (0.1 sec versus 10/3 seconds). Their PMSFF algorithm outperformed the nineteen methods tested based on F1-score and AUPRC for the NSP448 dataset. For subset NSP355, the PMSFF generally outperformed other methods and was slightly worse compared to D-PPisite (0.389 vs 0.390).  The authors compared their methods using subsets of data that focused on small ligands (S), nucleotides (N), heterodimers (He), homodimers (Ho), paratopes (Pa), and epitopes (Ep). Again, the PMSFF generally outperformed current methods, except for small ligand binding sites where PIPENN marginally was better. Limitations of their algorithm, include the lack of protein partner, nor structural information in their predictions.

Significance of work:

Overall, the significance of their methodology is the application of multi-sliding windows to capture fine detail and multi-kernel sizes combined with a bidirectional GRU to analyze PBRs on a global context.  

Minor edits

11)      The x-axis labels for the figures are low resolution and difficult to read

22)      Figure 3A & 3B colors for DeepProBind and PMSFF make it harder to see the vertical line inside the violin plot.

33)      Line 462 should read “the number of windows“ and NOT “the number the windows”.

44)      The following references are missing hyperlinks for DOI (reference # 56, 57, 66, 67, 83, 87, 90, 91)

Comments on the Quality of English Language

The manuscript is well written and there are only a few grammatical mistakes. 

Author Response

[Comment 1] The x-axis labels for the figures are low resolution and difficult to read.

[Response 1] Thank you for pointing out this. We regenerate all the figures to improve the resolution.

[Comment 2] Figure 3A & 3B colors for DeepProBind and PMSFF make it harder to see the vertical line inside the violin plot.

[Response 2] Thanks for your suggestion. We regenerate Figure 3 and change the color for PMSFF to distinguish with DeepProBind.

[Comment 3] Line 462 should read “the number of windows“ and NOT “the number the windows”.

[Response 3] Thank you for pointing out this. We check and correct this writing error throughout in this revised manuscript. “the number of windows” on page 16 (line 459) is corrected to “the number of windows” which is in blue color.

[Comment 4] The following references are missing hyperlinks for DOI (reference # 56, 57, 66, 67, 83, 87, 90, 91)

[Response 4] Thanks for pointing out this. We check the references and add all missing hyperlinks for DOI except reference 27 because we can’t find it.

Reviewer 2 Report

Comments and Suggestions for Authors

PMSFF Manuscript Review:

                  This manuscript describes the development of a new algorithm for predicting amino acids that are involved in binding interactions. The authors proposed method, called Prediction task through Multi-scale Sequence-based Feature Fusion (PMSFF), employs a sliding window scale that uses different size windows and multi-scale residue embeddings to incorporate the information from neighboring residues more effectively to provide more accurate predictions.  The authors do a rigorous comparison to other approaches to demonstrate that this approach in general provides more accurate predictions, though not in all use cases that were tested. This approach represents an improvement over existing methods and in general, the authors do not oversell this as a major step forward. They also mention that the major limitation in all current methods is the lack of consideration of the binding partner, which will be a more significant advance in the future. Overall, this is a good manuscript that is worthy of publication with a few small revisions.

Major critiques:

1.        There are a lot of acronyms in this field and many of them are used without definition the first time they are used.  For example, there are five deep learning models described on lines 36-37 that are not defined.  These may be very obvious to people in the field, but it will make the manuscript much more accessible if all abbreviations and acronyms are defined just to help the reader to look them up if they are not familiar with them.

2.        The first two paragraphs on page 2 (lines 38 – 68) could be made much clearer. The authors do a very thorough job of listing the major sequence and structure-based methods for predicting binding residues, but it reads like a list since each sentence starts with the author of the method paper and the method used in that approach.  It would be possible to revise this to read in a more engaging way that combined some of the similar methods or similar aspects from methods together. Again, this is sort of stylistic, but it would make the paper more accessible to a broader audience who may not have the same level of familiarity with the methods as the author.

Minor Critiques:

1.        There are several typos or grammatically incorrect statement throughout the manuscript.  Examples include:

Line 44 – “with attention creating” is not accurate in the current context.  I think the authors mean “to create” but I am not sure.

Line 67 – it should be “focused” and not “focus”

Line 121 – there should be a “the” in front of NSP6373

A careful read of the manuscript will ensure there are no additional phases or misspellings that need to be corrected.

Comments on the Quality of English Language

In general, the English is good but, as noted above, there are some incorrect tense for some terms and a few inaccurate phrases.

Author Response

[Comment 1] There are a lot of acronyms in this field and many of them are used without definition the first time they are used. For example, there are five deep learning models described on lines 36-37 that are not defined. These may be very obvious to people in the field, but it will make the manuscript much more accessible if all abbreviations and acronyms are defined just to help the reader to look them up if they are not familiar with them.

[Response 1] Thank you for pointing out this. We check all acronyms throughout in this revised manuscript and add the full spelling when first time used such as Multilayer Perceptron (MLP), Support Vector Machine (SVM), Random Forest (RF), Convolutional Neural Networks (CNN), and Recurrent Neural Network (RNN) which locates on page 2 (lines 43-45) in brown color and Graph Convolutional Network (GCN) which locates on page 2 (line 56) in brown color.

[Comment 2] The first two paragraphs on page 2 (lines 38 – 68) could be made much clearer. The authors do a very thorough job of listing the major sequence and structure-based methods for predicting binding residues, but it reads like a list since each sentence starts with the author of the method paper and the method used in that approach. It would be possible to revise this to read in a more engaging way that combined some of the similar methods or similar aspects from methods together. Again, this is sort of stylistic, but it would make the paper more accessible to a broader audience who may not have the same level of familiarity with the methods as the author.

[Response 2] Thanks for your suggestion. We reorganize the summary of sequenced-based and structured-based PPIs prediction methods in which we stress the novel features used of sequence-based approaches and different models used in structure-based approaches. And, the summary in this revised manuscript is shown on page 2 (lines 47-52 and lines 57-64) in blue color.

[Comment 3] There are several typos or grammatically incorrect statement throughout the manuscript. Examples include:

Line 44 – “with attention creating” is not accurate in the current context.  I think the authors mean “to create” but I am not sure.

Line 67 – it should be “focused” and not “focus”

Line 121 – there should be a “the” in front of NSP6373

A careful read of the manuscript will ensure there are no additional phases or misspellings that need to be corrected.

[Response 3] Thank you for pointing out this. We check and correct this writing error throughout in this revised manuscript. For example, we correct “creating” as “to create” on page 2 (line 51) in red color. We delete “focus” as we reorganize the summary of sequenced-based and structured-based PPIs prediction methods. And, we add “the” before “NSP6673” on page 5 (line 117) in red color.

Reviewer 3 Report

Comments and Suggestions for Authors

The authors present a novel, extensible method that uses only sequence data to predict which residues in a protein bind a small molecule, nucleotide or other protein, so-called protein binding residues (or PBRs). This paper represents an important albeit incremental improvement in PBR prediction.

This paper is very thorough, if anything, it could stand to be shortened. Nevertheless, it does not address the impact of improved structure prediction on PBR prediction. For example, the following papers report methods that now (can) incorporate structure predictions

McGreig, Jake E., et al. "3DLigandSite: structure-based prediction of protein–ligand binding sites." Nucleic acids research 50.W1 (2022): W13-W20.

Jakubec, David, et al. "PrankWeb 3: accelerated ligand-binding site predictions for experimental and modelled protein structures." Nucleic Acids Research 50.W1 (2022): W593-W597.

The authors may wish to revise the introduction to be a tad more succinct yet provide an even broader overview of the latest advances in PBR prediction.

Similarly, the results need to be a bit more focused. The authors have thoroughly evaluated their method, but do not present any specific examples to illustrate the strengths of their method. The authors should consider possibly placing some of their evaluations in an appendix (summarizing them in the results section) and including 1-3 examples, where their method successfully predicts PBRs, that illustrate the strengths (and possibly the limitations) of their approach.

In anticipation of the authors changing much of their text during the revision process, I have not provided too many specific corrections, but some do stand out to illustrate, in addition to the overall changes suggested above, specific revisions that would improve this paper. 

Line 1: The term "protein binding residues" confusing. Readers not familiar with the field may think that such residues are those predicted to be involved in protein/protein interactions (i.e. residues binding proteins) rather than residues in a protein which are likely to bind another molecule (macro or small). This is made clear on line 34f, but you probably should add a parenthetical note in the abstract. For instance, line 1 could read "Accurate prediction of protein binding residues (PBRs) - residues in a protein sequence that bind small or macro-molecules - is crucial for understanding cellular".

The legend for Figure 4 does not correspond with the figure. This figure seems to show AUROC scores for four different methods as applied to predicting PBRs binding small molecules, nucleotides, etc. The legend refers only to paratope and epitope prediction and indicates panels labeled A and B (which labels do not appear in the figure).

Comments on the Quality of English Language

Again, as I anticipate substantial revisions to the text, I am not providing a comprehensive list of changes required, but the following stand out:

Line 7f: "In this study, we propose one novel framework that could apply multiple types of PBRs Prediciton task through Multi-scale Sequence-based Feature Fusion (PMSFF) strategy" has a spelling error and could generally be improved. It could be changed, for example, to read "In this study, we propose a novel framework applicable to multiple types of PBRs: Prediction task through Multi-scale Sequence-based Feature Fusion (PMSFF)." The authors should also consider tightening up the abstract so it is a single, coherent paragraph.

Line 19f: "Series of cellular functions" does not quite make sense. Perhaps you mean "A series of cellular functions" or even "Multitudinous cellular functions"?

Author Response

[Comment 1] This paper is very thorough, if anything, it could stand to be shortened. Nevertheless, it does not address the impact of improved structure prediction on PBR prediction. For example, the following papers report methods that now (can) incorporate structure predictions.

McGreig, Jake E., et al. "3DLigandSite: structure-based prediction of protein–ligand binding sites." Nucleic acids research 50.W1 (2022): W13-W20.

Jakubec, David, et al. "PrankWeb 3: accelerated ligand-binding site predictions for experimental and modelled protein structures." Nucleic Acids Research 50.W1 (2022): W593-W597.

[Response 1] Thank you for pointing out this. In this revised manuscript, we address the impact of improved structure prediction on PBRs prediction which locates on page 2 (lines 35-40) in blue color.

[Comment 2] The authors may wish to revise the introduction to be a tad more succinct yet provide an even broader overview of the latest advances in PBR prediction.

Similarly, the results need to be a bit more focused. The authors have thoroughly evaluated their method, but do not present any specific examples to illustrate the strengths of their method. The authors should consider possibly placing some of their evaluations in an appendix (summarizing them in the results section) and including 1-3 examples, where their method successfully predicts PBRs, that illustrate the strengths (and possibly the limitations) of their approach.

[Response 2] Thanks for your suggestion. In this revised manuscript, we make the introduction more succinct through reorganizing the summary of related PPIs prediction methods (page 2 lines 47-52 and lines 57-64) in blue color.

We show the prediction results of a representative protein from the NSP448 set in the Supplementary File and illustrate the strengths of PMSFF on page 12 (lines342-344) in blue color.

[Comment 3] In anticipation of the authors changing much of their text during the revision process, I have not provided too many specific corrections, but some do stand out to illustrate, in addition to the overall changes suggested above, specific revisions that would improve this paper.

Line 1: The term "protein binding residues" confusing. Readers not familiar with the field may think that such residues are those predicted to be involved in protein/protein interactions (i.e. residues binding proteins) rather than residues in a protein which are likely to bind another molecule (macro or small). This is made clear on line 34f, but you probably should add a parenthetical note in the abstract. For instance, line 1 could read "Accurate prediction of protein binding residues (PBRs) - residues in a protein sequence that bind small or macro-molecules - is crucial for understanding cellular".

[Response 3] Thanks for your suggestion. We future explain PBRs in the begain of Abstract shown as follows: "Protein performs different biological functions through binding with various molecules which is mediated by a few key residues." which locates on Page 1 (lines 1-2) in blue color. This illustrates the meaning of PBRs that are residues involved in protein-protein interactions.

[Comment 4] The legend for Figure 4 does not correspond with the figure. This figure seems to show AUROC scores for four different methods as applied to predicting PBRs binding small molecules, nucleotides, etc. The legend refers only to paratope and epitope prediction and indicates panels labeled A and B (which labels do not appear in the figure).

[Response 4] Thank you for pointing out this. We correct the legend for Figure 4 in this revised manuscript shown as follows: “Comparison of PMSFF and other methods on type-specific PBRs. N and S mean the protein-Nucleotide (DNA, RNA) binding residues and protein-Small ligand binding residues. He and Ho stand for the binding residues from heterodimers and homodimers. Pa and Ep are short for paratope and epitope which are binding residues from antibody and antigen interaction interface, respectively.” which locates on page 14 in blue color.

Round 2

Reviewer 3 Report

Comments and Suggestions for Authors

The authors have improved this paper greatly and I thank them for the hard work they put into their revisions.

The only substantive critique I have at this point is that in the supplementary figure, there is a portion of what looks to be part of the actual binding site that is labeled as "false positive" (blue) in the DELPHI panel. Is this actually a miscoded false negative or just an artifact of the viewing angle (there is a nearby 'island' in the binding site that is not actually part of the binding site, but which DELPHI predicts to be in the binding site). If this is a miscoding, it needs to be fixed. If this is an artifact of the viewing angle, it needs to be explained.

Additionally, there are a few changes to the English language usage that need to be made (see that section of my review).

Comments on the Quality of English Language

Due to a technical glitch, it looks to me like my previous comments on the English language usage did not make it into the review you saw. From what I can tell, most of the issues I found are fixed in this revision, but at least one remains. In addition, there are a couple of edits required to the revisions you made.

Lines 1-2: should be revised -- pluralizing "protein" and linking the first two sentences" to read "Proteins perform different biological functions through binding with various molecules, which are mediated by a few key residues. Accurate prediction of such protein binding residues (PBRs)".

The beginning of the body of the paper reads "Series of cellular functions", which is rather awkward phrasing. Please change it to "Multitudinous cellular functions" or something like that.

The sentence starting on line 343 and continuing to the following line, "It can be seen from Fig.S1 that PMSFF predicts the most true interaction sites compared with other 13 methods." needs revision. This makes it sound like you are showing more than one example. Perhaps "As illustrated by the example shown in Fig.S1, PMSFF accurately predicts a greater portion of a typical PBR than 13 other commonly used methods." is better?

Author Response

[Comment 1] The only substantive critique I have at this point is that in the supplementary figure, there is a portion of what looks to be part of the actual binding site that is labeled as "false positive" (blue) in the DELPHI panel. Is this actually a miscoded false negative or just an artifact of the viewing angle (there is a nearby 'island' in the binding site that is not actually part of the binding site, but which DELPHI predicts to be in the binding site). If this is a miscoding, it needs to be fixed. If this is an artifact of the viewing angle, it needs to be explained.

[Response 1] Thank you for pointing out this. We checked the original file drawing the DELPHI panel and found that the 22th residue is wrongly labeled in blue color. In this revise version, we correct the 22th residue in red color (false negative) because DELPHI incorrectly predicts it as non-binding. Thanks again for pointing this out and we check all figure panels making sure there is no error like this.

[Comment 2] Additionally, there are a few changes to the English language usage that need to be made (see that section of my review).

Due to a technical glitch, it looks to me like my previous comments on the English language usage did not make it into the review you saw. From what I can tell, most of the issues I found are fixed in this revision, but at least one remains. In addition, there are a couple of edits required to the revisions you made.

Lines 1-2: should be revised -- pluralizing "protein" and linking the first two sentences" to read "Proteins perform different biological functions through binding with various molecules, which are mediated by a few key residues. Accurate prediction of such protein binding residues (PBRs)".

The beginning of the body of the paper reads "Series of cellular functions", which is rather awkward phrasing. Please change it to "Multitudinous cellular functions" or something like that.

The sentence starting on line 343 and continuing to the following line, "It can be seen from Fig.S1 that PMSFF predicts the most true interaction sites compared with other 13 methods." needs revision. This makes it sound like you are showing more than one example. Perhaps "As illustrated by the example shown in Fig.S1, PMSFF accurately predicts a greater portion of a typical PBR than 13 other commonly used methods." is better?

[Response 2] Thanks for your suggestions. We tried our best to improve the manuscript and made some changes to the manuscript. These changes will not influence the content and framework of the paper. And here we did not list the changes but marked in blue in the revised version. We appreciate for reviewer’s warm work eamestly and hope that the correction will meet with approval.